# New Benzo[*h*]quinolin-10-ol Derivatives as Co-sensitizers for DSSCs

**DOI:** 10.3390/ma14123386

**Published:** 2021-06-18

**Authors:** Aneta Slodek, Paweł Gnida, Jan Grzegorz Małecki, Grażyna Szafraniec-Gorol, Pavel Chulkin, Marharyta Vasylieva, Jacek Nycz, Marcin Libera, Ewa Schab-Balcerzak

**Affiliations:** 1Institute of Chemistry, University of Silesia, 9 Szkolna Str., 40-006 Katowice, Poland; a.slodek@wp.pl or aneta.slodek@us.edu.pl (A.S.); jan.malecki@us.edu.pl (J.G.M.); grazyna.szafraniec-gorol@us.edu.pl (G.S.-G.); , marcin.libera@us.edu.pl (M.L.); 2Centre of Polymer and Carbon Materials, Polish Academy of Sciences, 34 M. Curie-Skłodowska Str., 41-819 Zabrze, Poland; pgnida@cmpw-pan.edu.pl (P.G.); mvasylieva@cmpw-pan.edu.pl (M.V.); 3Faculty of Chemistry, Silesian University of Technology, 9 Strzody Str., 44-100 Gliwice, Poland; pavel.chulkin@polsl.pl

**Keywords:** DSSC, 2-cyanoacrylic acid derivatives, co-adsorbents, EIS

## Abstract

New benzo[*h*]quinolin-10-ol derivatives with one or two 2-cyanoacrylic acid units were synthesized with a good yield in a one-step condensation reaction. Chemical structure and purity were confirmed using NMR spectroscopy and elemental analysis, respectively. The investigation of their thermal, electrochemical and optical properties was carried out based on differential scanning calorimetry, cyclic voltammetry, electronic absorption and photoluminescence measurements. The analysis of the optical, electrochemical and properties was supported by density functional theory studies. The synthesized molecules were applied in dye-sensitized solar cells as sensitizers and co-sensitizers with commercial N719. The thickness and surface morphology of prepared photoanodes was studied using optical, scanning electron and atomic force microscopes. Due to the utilization of benzo[h]quinolin-10-ol derivatives as co-sensitizers, the better photovoltaic performance of fabricated devices compared to a reference cell based on a neat N719 was demonstrated. Additionally, the effect of co-adsorbent chemical structure (cholic acid, deoxycholic acid and chenodeoxycholic acid) on DSSC efficiency was explained based on the density functional theory.

## 1. Introduction

Much attention is currently being paid to renewable energy sources. This is related to increasing environmental pollution, the gradual depletion of fossil resources and environmental degradation. One of the most promising renewable energy sources is solar energy, which can be used for heating or converted into electricity [1]. It is worth paying particular attention to photovoltaics (PV), which have been widely and intensively developed in recent years [2,3,4]. In addition to well-known inorganic materials, organic materials are intensively tested in PV devices. Increased research on third generation PV cells accompanied the discovery and development of the first dye-sensitized solar cell (DSSC) in 1991 [5]. However, the DSSCs became more popular around the year 2000 when their efficiency exceeded 10% [6]. Additionally, they are gaining popularity and are constantly being researched due to their economic advantages: low cost of fabrication, high power conversion efficiency, and relatively straightforward procedures of manufacturing [5,7]. It is worth noting that the DSSC cell consists of several key elements, including the dye used. The dye is responsible for absorbing incident photons and injecting electrons into the TiO_2_ conduction band [8]. Currently, metal-containing complexes such as N3, N719, N749 based on Ru or YD2 with Zn are mainly utilized to prepare dye-sensitized solar cells [9,10,11,12]. However, metal-containing dyes are expensive to obtain and the purification process is very complicated and tedious [13,14,15,16]. Therefore, research is being carried out to reduce the use of sensitizers containing metal atoms by looking for new metal-free dyes that are cheaper and easier to synthesize. Organic sensitizers are used in exchange for metal-dyes or as a co-sensitizer to reduce the amount of commercial dyes used. Very important at the design level are the properties that the new dye should exhibit, including good solubility in solvents, wide absorption range of sunlight, high molar coefficient, resistance to photodegradation and photocorrosion, thermal and electrochemical stability, and appropriate alignment of the frontier molecular orbital energy levels [17,18,19,20,21,22]. It was found that co-sensitization using two or more sensitizers is an effective approach to achieve highlight harvesting efficiency [23,24]. Two groups of DSSC can be specified. The first one is based on two dyes, one of which is a new metal-free dye, and the other is a commercial metal-containing sensitizer, which is often Ru (e.g., N719, N3, HD-2) [7,23,24,25,26,27,28,29,30,31,32]. It is worth noting that these PV cells exhibit higher efficiencies than devices containing only commercial dye in most cases. Even if the efficiencies are comparable or slightly lower than those of the reference DSSC, a correspondingly less amount of metal-dye amount is used, which reduces the cost of preparing the solar cell [33,34,35]. The second group constitutes DSSC, sensitized only, with new metal-free dyes [33,35,36,37,38,39,40,41].

Herein, two new benzo[*h*]quinolin-10-ol cyanoacrylic acid derivatives are presented for use as co-sensitizers with commercial N719 dye in DSSC devices. The properties of new compounds based on thermal, UV-vis, photoluminescence, and cyclic voltammetry measurements are determined. Additionally, functional density theory (DFT) estimates the frontier molecular orbitals of the synthesized molecules, UV-vis spectra adsorbed on TiO_2_, and binding energies of the dyes on TiO_2_ applied. Moreover, the impact of the chemical structure of typically used adsorbents, such as chenodeoxycholic acid (CDCA), cholic acid (CA), and deoxycholic acid (DCA) on device performance based on N719 was explained by DFT. The fabricated devices were characterized by the current density-voltage (J-V) curves and electrochemical impedance spectroscopy (EIS). It was demonstrated that the new simple molecules, applied as co-sensitizer with N719, allowed us to obtain higher or comparable efficiencies to reference cells based only on expensive N719 dye.

## 2. Experimental

All chemicals were commercially available and were used without further purification. 10-Hydroxybenzo[*h*]quinoline-9-carboxaldehyde and 10-hydroxybenzo[*h*]quinoline-7,9-dicarboxaldehyde were synthesized, as has been reported [42]. The chemicals, instrumental equipment applied for synthesized compounds and solar cells characterization and DSSC preparations are presented in Appendix A.

### 2.1. Computational Details

All theoretical calculations were performed using Gaussian 16, Revision C.01, program package [43] at DFT or TD-DFT level. The singlet state geometry optimizations, frequency and electronic transition calculations were made with the use of B3LYP functional [44,45] with the 6-31G(d,p) basis set [46]. Calculations were made in the gas phase except for the absorption electronic spectra of 1a, 2a compounds (Appendix A) in addition to the PCM model [47], with DMF as a solvent. The density of states diagrams (Appendix A) was obtained with GaussSum [48]. The adsorption energies (E_ads_) of the dyes were evaluated using the following expression, E_ads_ = E_dye_ + E_TiO2_ − E_dye_@TiO_2_, where E_dye_, E_TiO2_ and E_dye_@TiO_2_ are the energies of dye, TiO_2_, and total system (dye@TiO_2_). The Gibbs free energies of the dye@TiO_2_ species were calculated at 298.15 K.

### 2.2. Synthesis of Cyanoacrylic Acids 10-Hydroxobenzo[h]Quinoline Derivatives

Carboxaldehyde and cyanoacetic acid (6 eq. for compound **1**; 16 eq. for compound **2**) were loaded into a Schlenk flask equipped with a magnetic stir bar. The flask was evacuated and flushed with argon for three cycles. Next, a mixture of MeCN/CHCl_3_ (3:1) was added, and the mixture was again evacuated and flushed with argon. Then, piperidine (12 eq. for compound **1**; 22 eq. for compound **2**) was added, and finally, the flask was flushed with argon and evacuated. The mixture was heated at 90 °C for 24 h. After being cooled to room temperature, the mixture was evaporated under reduced pressure, water was added, and the crude product was extracted with CH_2_Cl_2_. The organic layer was collected and dried over anhydrous MgSO_4_. After filtration, the solvent was removed under reduced pressure. The residue was purified by crystallization from a mixture of EtOH/Et_2_O.

10-Hydroxybenzo[*h*]quinoline-9-cyanoacrylic acid (**1a**) Yield: 54%. ^1^H NMR (400 MHz, DMSO) δ 9.04 (d, *J* = 4.4 Hz, 1H), 8.66 (m, 1H), 8.61 (d, *J* = 8.0 Hz, 1H), 8.19 (d, *J* = 8.4 Hz, 1H), 8.10–8.03 (d, *J* = 8.4 Hz, 1H), 7.99–7.92 (m, 1H), 7.83 (m, 2H). ^13^C NMR (101 MHz, DMSO) δ 165.20, 162.17, 146.40, 142.74, 138.59, 134.00, 128.97, 128.16, 126.29, 125.44, 122.28, 120.56, 119.88, 118.66, 117.85, 114.99, 114.88, 111.13. Elem. Anal. (%) Calcd for C_17_H_10_N_2_O_3_: C, 70.34; H, 3.47; N, 9.65. Found: C, 70.18; H, 3.77; N, 9.84.

10-Hydroxybenzo[*h*]quinoline-7,9-di(cyanoacrylic) acid (**2a**): Yield: 60%. ^1^H NMR (400 MHz, DMSO) δ 9.12 (d, *J* = 4.4 Hz, 1H), 8.91 (s, 1H), 8.76 (d, *J* = 8.0 Hz, 1H), 8.60 (d, *J* = 1.8 Hz, 2H), 8.17 (d, *J* = 8.8 Hz, 1H), 8.07 (d, *J* = 8.8 Hz, 1H), 7.94–7.91 (m, 1H). ^13^C NMR (101 MHz, DMSO) δ 165.01, 164.63, 162.14, 146.86, 146.68, 142.74, 138.56, 134.80, 128.97, 128.18, 126.29, 124.22, 123.17, 120.52, 119.17, 118.65, 117.49, 114.99, 114.94, 114.86, 111.42. Elem. Anal. (%) Calcd for C_21_H_11_N_3_O_3_: C, 65.46; H, 2.877; N, 10.905. Found: C, 65.06; H, 2.93; N, 10.41.

## 3. Results and Discussion

### 3.1. Synthesis, Thermal, Optical and Electrochemical Characterization

The preparation route and chemical structure of novel compounds 10-hydroxybenzo[*h*]quinoline-9-cyanoacrylic acid and 10-hydroxybenzo[*h*]quinoline-7,9-di(cyanoacrylic) acid, denoted as **1a** and **2a**, respectively, are presented in Figure 1. The synthesis of the cyanoacrylic acids 10-hydroxobenzo[*h*]quinoline derivatives (**1a** and **2a**) was carried out by the Knoevenagel condensation reaction of **1** and **2** in the presence of cyanoacetic acid and piperidine. The aldehyde precursors **1** and **2** were prepared according to the previously described procedure [42].

The desired dyes **1a** and **2a** were obtained in good yield (54–60%) as an orange (**1a**) and red solid (**2a**). The ^1^H and ^13^C NMR spectroscopic characterization and elemental analysis confirmed its chemical structure and purity. DSC thermograms registered in the first heating scan showed the melting endotherms with a maximum at 240 and 175 °C, for **1a** and **2a**, respectively (cf. Appendix A). The second heating run, recorded after cooling, revealed only the glass transition (T_g_) temperature at 202 (**1a**) and 119 °C (**2a**), and during further heating, no additional peaks were seen. It shows that the prepared compounds are molecular glasses, which form a stable glassy phase. The presence of a second cyanoacrylic acid group lowered the melting point, but conversely, slightly impacted T_g_.

The electronic absorption and photoluminescence (PL) spectra of the cyanoacrylic acids 10-hydroxobenzo[*h*] quinoline derivatives in methanol (MeOH) and dimethylformamide (DMF) are shown in Figure 2, and the data are listed in Table 1. Additionally, the UV-vis data of starting aldehydes (**1** and **2**) are given in Appendix A and Appendix A.

The synthesized compounds (**1a** and **2a**) reveal two absorption bands, with one of them assigned to the π-π* electronic excitations localized within the conjugated system, ranging from 260 to 360 nm, and with a maximum of 278 (**1a**), 280 (**2a**) nm and shoulder of 328 (**1a**), 354 nm (**2a**), respectively. The second is seen at a lower energy range with λ_max_ at 391 (**1a**) and 407 (**2a**), which can be attributed to the intramolecular charge transfer (ICT) from 10-hydroxybenzo[*h*]quinoline to the electron withdrawing anchoring fragment. The λ_max_ of ICT of **2a** is bathochromically shifted (16 nm), and the band is broader and in a visible range compared to **1a**. Usually, the increase in cyanoacrylic acid groups shifts the absorption range to a lower energy [23,25]. In this case, the effect of the number of cyanoacrylic acid units on the electronic absorption range is weakly pronounced. A good match between the experimental and obtained from DFT calculations absorption spectra was observed (cf. Figure 3). Compound **2a**, with two anchoring groups, exhibits a higher molar absorption coefficient (ε = 10,667 at 354 nm and 10,000 M^−1^cm^−1^ at 407 nm) than compound **1a** with one cyanoacrylic acid moiety (ε = 6000 at 328 nm and 6667 M^−1^cm^−1^ at 391 nm) in DMF. Thus, a higher molar absorption coefficient and red-shifted spectrum of **2a** can be advantageous for a higher DSSC efficiency.

When comparing the UV-vis spectra of **1a** and **2a** with the absorption of aldehydes, it was found that the replacement of CHO units in **1** and **2** by the cyanoacrylic acid group in **1a** and **2a** results in a bathochromic shift of 10 nm in the ICT peak in the case of **1a**, while for **2a**, the position of ICT remains unchanged (Table 1 and Appendix A). The absorption shapes of **1** and **1a** are similar, whereas in the case of **2a**, the S_0_→S_2_ transition is weakly expressed when compared to compound **2**. Conversely, compounds **1a** (ε_max_ = 5085 M^−1^cm^−1^) and **2a** (ε_max_ = 12,727 M^−1^cm^−1^) have a considerably higher molar extinction coefficient for the S_0_→S_1_ transition compared to both aldehyde derivatives **1** (ε_max_ = 4505 M^−1^cm^−1^) and **2** (ε_max_ = 208 M^−1^cm^−1^) in methanol solution.

Next, the absorption nature of synthesized compounds adsorbed into TiO_2_ was tested. Figure 2a displays the UV-vis spectra of a **1a** and **2a** measured for TiO_2_ films with adsorbed dyes. In the UV-vis spectra of **1a** and TiO_2_ system (**1a**@TiO_2_) one band with a maximum at 406 nm was observed. In the case of **2a**@TiO_2_ band, two structured maxima at 350 and 395 nm is seen. When the solution state and TiO_2_ systems spectra were compared, both adsorbed molecules shown shifted absorption, which is likely due to aggregation [49,50]. Moreover, TiO_2_ with dyes exhibited an extended absorption curve up to 600 nm, which was about 150 nm higher than the solution state absorption curve, probably due to the J-aggregates’ formation [49,50]. Considering the UV-vis spectra of TiO_2_ with dyes, the broader absorption covering the range from 300 to 600 nm exhibited **2a**@TiO_2_ compare to **1a**@TiO_2_ which absorbed radiation from 340 to 600 nm (cf. Figure 2). The effect of chenodeoxycholic acid (CDCA) as co-adsorbent on UV-vis properties of the prepared photoanodes was also studied. Some of the most commonly used co-adsorbents for DSSC are cholic acid (CA), deoxycholic acid (DCA) and chenodeoxycholic acid (CDCA). In this article, CDCA was selected based on our previous results [51], and the DFT calculations described herein in Section 3.2.1. The expected decrease in absorbance of the photoanode containing CDCA was seen (cf. Figure 2b). The working principle of the co-adsorbent is based on the adsorption of CDCA particles instead of dye molecules, which reduces the formation of dyes aggregates. Due to the smaller number of dye molecules anchored to the TiO_2_ surface, the absorbance decreases.

In PL spectra of **1a** and **2a**, two emission maxima in the range of 400–675 nm are seen. Similar to absorption spectra, a hypsochromic shift in emission spectra was observed for bi anchoring **2a** compared to mono anchoring **1a**. Compounds **1a** and **2a** demonstrate very weak fluorescence in both solutions with the PL quantum yields range from 0.07 to 1.25% higher for bi anchoring dye **2a** (Table 1). A comparable increase in quantum efficiency was observed for the pyrazolo[3,4-*b*]quinoline and phenothiazine derivatives when increasing the number and/or strength of electron withdrawing substituents [52,53]. Compounds **1a** and **2a** possess bixponential fluorescence decay in both solvents (Table 1 and Appendix A). The efficient lifetimes of **1a** and **2a** are comparable in the same solvent and slightly shorter in DMF (4.24–4.57 ns) than in MeOH (5.56–5.97 ns). The time-resolved data and emission range imply that for dyes **1a** and **2a**, the ICT character is lessened, and the fluorescence may originate from π→π^*^ excited state because ICT is a competitive relaxation process of the singlet excited state and typically reduces the fluorescence [54].

The cyclic voltammetry (CV) measurements were carried out to study the electrochemical behavior of the synthesized benzo[*h*]quinolin-10-ol derivatives and N719 in Bu_4_NPF_6_/DMF solution. During the CV experiment, the reduction process was not observed, and oxidation processes for **2a** and N719 were seen (Appendix A). The oxidation potential onset (E_ox_^onset^) of **2a** was higher (0.53 V) compared to N719 (0.33 V). A slightly higher oxidation potential (0.35 V) of N719 was determined from DPV measurements (Appendix A). Comparing the structure of **2a** and **1a**, it is expected that the oxidation potential of **2a** (with two electron withdrawing groups) should be lower than **1a** [54]. It can be assumed that the oxidation potential of **1a** is higher than 0.53 V (**2a**). Unfortunately, it is impossible to measure the higher potential in DMF. Changing the solvent was not considered, due to the limited solubility of these compounds. Based on the oxidation onset, the potential HOMO energy level was calculated assuming that the IP of ferrocene equals −5.1 eV [55], being −5.73 and −5.63 eV for **2a** and N719, respectively. HOMO levels are somewhat more negative than the I^−^/I^3−^ redox potential (−4.8 eV) [32], and should enable the dye regeneration. The LUMO energy level determined from the optical energy gap (Eg^OPT^), and calculated HOMO to be −3.03 and −3.47 eV for 2a and N719, respectively. A HOMO energy level of N719 was reported in the literature from UPS (−5.34 eV vs vacuum) and CV in DMF (−5.37 eV vs vacuum); these measurements are similar to our experimental result [56,57]. However, the lower HOMO energy level in the range of 6.0–6.14 eV obtained in different CV experimental conditions was also reported [58,59,60]. Additionally, CV measurements of N179 were carried out in acetonitrile (ACN) and DMF with various scanning rates. In both cases, an irreversible oxidation process was demonstrated (Appendix A, Appendix A). At different scan rates, the beginning of the peak did not change.

### 3.2. Computational Studies

Density functional theory (DFT) studies were applied for (i) explanation of the co-adsorbents chemical structure importance, (ii) both neat compounds and dye-TiO_2_ system geometry optimizations and frontier molecular orbitals energy determination of **1a** and **2a,** and (iii) UV-vis absorption spectra calculations of dyes in solution and dye-TiO_2_ systems.

#### 3.2.1. Importance of Co-Adsorbents Chemical Structure

The co-adsorbents are applied for DSSC photoanode preparation to limit the dye aggregates formation onto TiO_2_ surface which may improve the photovoltaic performance of the device. As co-adsorbents, chenodeoxycholic acid (CDCA), cholic acid (CA) and deoxycholic acid (DCA) are usually used [61]. Among them, the beneficial impact of CDCA on DSSC efficiency was presented in our previous work [51]. Moreover, based on DFT, G. Saranya et al. [62] showed that CDCA co-adsorbent is a crucial component of a high-performance DSSC; they studied effect of CDCA on the properties of the dye denoted as TY6′ and TiO_2_ interface. It was found that CDCA not only stabilizes the TY6′/TiO_2_ system, but also prevents the surface tensile stress induced by the dye monolayer [62].

Considering the chemical structure of CDCA, CA, and DCA, they differ structurally only by their R substituents. There is a lack of an explanation of the effect of the chemical structure of CA, DCA, and CDCA on DSSC performance in the literature. Thus, to explain the impact of co-adsorbents CDCA, CA and DCA chemical structure, DFT was used. The geometric optimizations and frequency calculations were made with the use of functional B3LYP [44,45] with the smaller 6-31G(d) basis set [46]. The calculations were carried out with the Gaussian 16, Revision C.01 program [43]. The molecular geometry, HOMO and LUMO contours of the Ti_21_O_54_H_24_ cluster, CA, CDCA, DCA co-adsorbent, as well as the adsorbed *enol*–forms of the compounds on Ti_21_O_54_H_24_ cluster are given in Appendix A.

A TiO_2_ surface model was built by cutting two trilayers from the bulk experimental geometry (with Ti–O bond lengths of 1.948 Å and 1.981 Å) and closed by hydrogen atoms (with optimized positions) forming terminal hydroxyl groups, which led to a Ti_21_O_54_H_24_ cluster (Figure 3).

A two-trilayer thickness was used to allow fivefold coordinated Ti atoms on the surface. Toward the other two directions, the model’s size was chosen to have a central fivefold coordinated Ti atom on the surface with two neighbouring fivefold coordinated Ti atoms and two neighbouring bridging oxygen atoms.

Two types, monodentate through C=O and bidentate bridging, namely the *enol*– and *keto*– forms, of anchoring modes of the dyes were calculated. The Ti–O_carboxylate_ bond distances in *enol*–form (cf. Appendix A) of 2.23 Å (CA), and 2.18 Å (CDCA, DCA) are longer than in the case of *keto*—bidentate form, where the distances are 2.11, 2.14 Å in CA, 2.09, 2.14 in CDCA and 2.09 and 2.18 Å in DCA. Thus, the bidentate bridging mode creates a more stable system. This is especially evident in the case of CA dye, for which the calculated adsorption energy of *keto*–form is 12 kcal/mol higher compared to the *enol*– one. The adsorption energies, calculated using the expression *E_ads_* = *E_dye_* + *E_TiO_*_2_ − *E_dye_*_@*TiO*2_, where *E_dye_*, *E_TiO_*_2_ and *E_dye_*_@*TiO*2_ are the energies of dye, Ti_21_O_54_H_24_, and total system (dye@ Ti_21_O_54_H_24_) of the *enol*– forms of these dyes are similar to the differences, do not exceed 2.4 kcal/mol (cf. Figure 4 and Appendix A).

The *keto*–form of CA shows a higher adsorption energy by about 9 kcal/mol than CDCA and DCA dyes. Moreover, one of the hydroxyl groups of CA molecule forms a hydrogen bond with TiO_2_, which results in the slope of the carbon skeleton and prevents interaction of sensitizer and TiO_2_ surface (Figure 5).

The adsorbed CDCA molecule is almost perpendicular to the TiO_2_ surface. The slope of the carbon skeleton increases with CDA and reaches a significant value for CA (cf. Appendix A). Thus, the high adsorption energy, highest *G*_ads_ and the bent geometry of the carbon skeleton of the adsorbed CA molecule indicate the smallest suitability for use in cells among the dyes tested. The adsorption energies of CDCA and DCA molecules are similar, however, taking into account the almost perpendicular adsorption of the CDCA molecule and the slightly higher value of *G*_ads_, it can be assumed that CDCA is a better co-adsorbent than DCA.

#### 3.2.2. Dye and Dye-TiO_2_ System Geometry Optimizations and Frontier Molecular Orbitals Energy Determination

The geometry optimizations and frequency calculations were made with of the use the B3LYP functional [44,45] with the 6-31G(d,p) basis set [63,64]. The calculations were carried out with the Gaussian 16, Revision C.01 program [43]. Adsorbed **1a** and **2a** molecules on the clusters are presented in Figure 6.

To obtain the binding energies of the dyes on TiO_2_, the structures of **1a**, **2a** and Ti_21_O_54_H_24_ cluster were optimized separately (cf. Section 3.2.1), before the dye@Ti_21_O_54_H_24_ systems were optimized. During the optimization of the dye@Ti_21_O_54_H_24_ system, the carboxylic proton of the dye was transferred to a nearby oxygen atom of the TiO_2_. Hence, the dyes were adsorbed on TiO_2_ with a bidentate bridging mode. Due to the intimate contact between carboxylate anchor-based dyes and the metal oxide surface, the structures with bidentate bridging modes exhibit superior stability compared to other types of anchoring modes. Because of the presence of two carboxyl groups in the **2a** molecule, the geometries of two possible types (**2a-a** and **2a-b**) of adsorption of this compound on the TiO_2_ surface have been optimized. Adsorption energies calculated using the expression *E_ads_* = *E_dye_*_@*TiO*2_ – (*E_dye_* + *E_TiO_*_2_), where *E_dye_*, *E_TiO_*_2_ and *E_dye_*_@*TiO*2_ are the energies of dye, Ti_21_O_54_H_24_, and total system (dye@Ti_21_O_54_H_24_). The obtained binding energies were –42.11, –37.37 and –52,72 kcal/mol for **2a-a**, **2a-b** and **1a**, respectively. The Ti–O_carboxylate_ bond distances are of 2.16, 2.18 Ǻ in **2a-a**@Ti_21_O_54_H_24_, 2.14, 2.21 Ǻ in **2a-b**@Ti_21_O_54_H_24_ and in the case of **1a**@Ti_21_O_54_H_24_ the distances are 2.11 and 2.18 Ǻ. The higher adsorption energy, and especially the high value of Δ*G*_ads_, indicates that the interaction of **2a** with TiO_2_ takes place through the carboxylate group in position 7 of the benzoquinoline ring i.e., in the form presented in Figure 6 as Ti_21_O_54_H_24_⋯**2a-a**. Adsorption through the carbonyl group in the *trans* position to the hydroxyl group in the benzoquinoline ring is also strengthened by the direction of the molecule’s dipole moment, as shown in Appendix A.

In the case of both dyes, there is a possibility of the formation of hydrogen bonds between the dye molecule and the TiO_2_ surface (cf. Appendix A). Whereas in the **2a-a**@Ti_21_O_54_H_24_ system, a cyano group and a proton belonging to the hydrated TiO_2_ surface are involved in such a bond, of **2a-b**@Ti_21_O_54_H_24_ the hydroxyl group on the surface interacts with the nitrogen atom in the benzoquinoline ring. Two types of hydrogen bonds are possible in the **1a**@Ti_21_O_54_H_24_ system. One between the –C≡N group and hydrated titanium oxide, and the second between the hydroxyl group at position 10 of the benzoquinoline ring and the oxygen of TiO_2_.

The geometry of the adsorbed molecules in the range of angles between the plane of the benzoquinoline ring and the carboxylate anchor group is significantly different from that of free dye molecules (cf. Appendix A). In adsorbed dye molecules, the plane of aromatic rings is twisted in relation to the plane of the 2-cyanoacrylic unit, and additionally in the **2a-b**@Ti_21_O_54_H_24_ and **1a**@Ti_21_O_54_H_24_ the benzoquinoline rings, which are tilted over the TiO_2_ surface. The relatively smallest changes occur in the thermodynamically and energetically privileged **2a-a**@Ti_21_O_54_H_24_ system, for which the angle between the benzoquinoline plane and the titanium dioxide surface does not differ much from the right angle (85.17°). Moreover, the internal geometry of the **2a** molecule adsorbed by carboxylate anchor in position 7 of benzoquinoline changes slightly, while the plane of the aromatic rings is twisted in relation to the plane of the carboxylate group only by about 5°.

The frontier molecular orbitals (FMO) of the dyes adsorbed on Ti_21_O_54_H_24_ system (c.f. Appendix A) reveal that HOMOs are localized in the dye while LUMOs have the shape of the lobes characteristic of *d*-type orbitals of titanium atoms. The analysis of the density of states diagrams (cf. Appendix A) shows that in **2a**@Ti_21_O_54_H_24_, the dye orbitals have a certain share (14%) of the HOMO level. In the case of the **1a**@Ti_21_O_54_H_24_ system, HOMO is located entirely on the dye (99%). Thus, the contribution of the dye is more significant in the valence band. LUMO in both systems is located on the Ti_21_O_54_H_24_.

The UV-vis spectra of the **2a**@Ti_21_O_54_H_24_ and **1a**@Ti_21_O_54_H_24_ were simulated in a vacuum using TF-DFT formalism depicted in Figure 7.

The transitions calculated in the visible range of the electronic absorption spectrum correspond to the excitations between the HOMO and the lowest energetically LUMO levels. Considering the location of the HOMO and LUMO orbitals (cf. Appendix A), it can be concluded that in the case of **1a**, the process of charge transfer from the dye molecule to TiO_2_ is more pronounced. This is due to the fact that the HOMO orbital in the **1a**@Ti_21_O_54_H_24_ system is located on the dye molecule, while in **2a**@Ti_21_O_54_H_24_ the proportion of the dye in HOMO is 14%. Thus, considering the calculation results concern FMO appear that **1a** could be a better sensitizer than **2a**, however, a wider range of absorption, the thermodynamics of adsorption as well as changes in the geometry of the adsorbed dyes indicate **2a-a** as a better sensitizer.

### 3.3. DSSCs Characterization

The synthesized cyanoacrylic acids 10-hydroxobenzo[*h*]quinoline derivatives (**1a** and **2a)** were applied as sensitizers and co-sensitizers in DSSCs. Photovoltaic devices with a FTO/TiO_2_ + dye/EL-HSE/Pt/FTO structure were fabricated. The three types of PV cells were prepared: (i) with a neat **1a** and **2a**, (ii) with a mixture of a new dye with the N719 and (iii) with a mixture of dyes and co-adsorbent. As co-adsorbent, based on our previous studies (described in [51]), and the DFT results presented in Section 3.2.1, chenodeoxycholic acid (CDCA) was selected. Moreover, the reference cell with N719 was constructed.

#### 3.3.1. Photoanode Thickness and Morphology

In the first step of the investigation, prepared photoanodes’ thickness and surface morphology were investigated using optical, scanning electron (SEM) and atomic force microscopes (AFM). The TiO_2_ layer thickness significantly impacts the photovoltaic parameters of the solar cells. The thicker layers can adsorb more dye molecules, but too much absorbed molecules may cause higher electron transport resistance and increase the recombination of electron with I_3_^−^ on the TiO_2_ surface, which decrease the open-circuit voltage (V_oc_) of the device and, consequently, lower the power conversion efficiency (PCE). Additionally, too thick an oxide layer may limit electron generation. Conversely, in thin TiO_2_ layers, not enough dye number can be anchored, causing a decrease in short-circuit current (J_sc_) values, which is reflected in PCE reduction [51,65,66]. The obtained TiO_2_ layers with adsorbed dyes molecules thickness, root-mean-square (RMS) determined based on AFM and root-mean-square roughness (S_q_) and sharpness of the roughness profile (S_ku_) obtained from optical microscope measurements are collected in Table 2.

The tested substrates with anchored dyes molecules showed thicknesses ranging from 8 to 12 µm. In the presented work, the thickness of the TiO_2_ was measured using two methods, optical and SEM microscopes. The advantage of the SEM over the optical microscope is that the thickness of individual layers can be determined. However, that thickness determination using an optical microscope is a non-destructive method. The representative cross-sectional SEM image and micrograph of photoanodes are depicted in Figure 8 and Appendix A. On the SEM images the layer of glass, FTO and mesoporous TiO_2_ were seen (cf. Figure 8 and Appendix A). Moreover, based on SEM, the presence of spherical TiO_2_ nanoparticles was confirmed. Considering the data given in Table 2, it can be concluded that only small differences in the obtained thickness values of TiO_2_ were observed using two independent measuring methods (SEM and optical microscope).

The values of surface quality parameters (RMS) determined by AFM were similar for most of the photoanodes tested. An increase in the roughness of the electrodes was observed when co-adsorbent was used. By analyzing the parameters S_q_ and S_ku_, it was possible to observe the trend of their change depending on the new dye used. In each of the three types of electrode structures, the mentioned parameters had lower values for compound **1a**. Moreover, similarly to the RMS values, the parameters described (S_q_ and S_ku_) increased after the addition of CDCA, which was due to lead to fewer dye molecules being anchored to the oxide substrate and thus less smoothing.

#### 3.3.2. Photovoltaic Properties of Devices

The photovoltaic (PV) properties of fabricated cells under standard AM 1.5 irradiations (100 mWcm^−2^) were analyzed using current-voltage (J–V) characteristics and the electrochemical impedance spectroscopy (EIS). Based on J–V plots, open-circuit voltage (V_oc_), photocurrent density (J_sc_), fill factor (FF), and power conversion efficiency (PCE) were calculated as the PV parameters (cf. Table 3). Photocurrent density−voltage curves (J−V) of fabricated devices are presented in Figure 9.

As can be expected considering the UV-vis absorption range of synthesized **1a** and **2a**, the devices based on neat cyanoacrylic acids 10-hydroxobenzo[*h*]quinoline derivatives exhibited significantly lower PV performance compare to a reference cell based on N719. This is mainly due to low J_sc_ values, which are largely related to the absorption properties of the dyes. This was due to the relatively narrow range and lower absorbance of the studied compounds compare to N719 (cf. Appendix A). The utilization of the synthesized compounds as co-sensitizers allowed the increase of the PCE up to 6.02 and 6.38% (**1a** + N719 and **2a** + N719, respectively) compared to reference cell based on a neat N719 (PCE = 5.35%). The higher efficiency of the dye-mixed devices is mainly attributed to improvement in J_sc_. The increase in J_sc_ was due to the extension of the absorption range of the photoanode containing a mixture of dyes, that is, N719 and cyanoacrylic acid 10-hydroxobenzo[*h*]quinoline derivatives compare to the absorption of the neat compounds (cf. Appendix A). The utilization of CDCA as co-adsorbent for photoanode preparation results in a further increase of PCE due to raising V_oc_, J_sc_ and FF. In the case of CDCA addition, only the cell containing molecules **2a** and N719 showed a higher efficiency (7.22%) than the reference cell (6.90%). The cell prepared with dye **1a** and N719 had lower photovoltaic parameters, including efficiency (6.42%). In each case, devices containing the dye **2a** showed better efficiency than a solar cell with compound **1a** due to mainly higher reached short-circuit current density. Photoanodes containing anchored molecules of dye **1a** and N719, both with and without the addition of CDCA, had higher absorption compared to photoanodes based on **2a** and N719.

According to the DFT calculations concerning FMO energy levels, a device with 1a showed better performance than 2a. However, electrochemical impedance spectroscopy studies have shown differences in the layer-by-layer resistances of these two types of solar cells favoring cells containing compound **2a**.

To study the intrinsic charge transfer properties of the cells, the electrochemical impedance spectroscopy (EIS) technique was used. Impedance spectroscopy method involves analysis of investigated system response under AC conditions in a range of frequencies. Each cell under light exposure of simulated solar light source of 1000 Lm/m^2^ was exposed to AC voltage signal. The DC voltage signal applied by a potentiostat was maintained at 0 V to reach short-circuit conditions. As each cell has its particular open-circuit voltage value, any particular non-zero voltage applied to different cells would result in electrodes’ different polarization. Therefore, the short-circuit conditions were chosen to measure impedance spectra as they allow us to compare the cells operating at the maximum current regime. The set of obtained impedance spectra are shown in Figure 10. Figure 10 gives a comparison between impedance spectra recorded in dark and under light illumination.

The complex plane plots (Figure 10) have a semicircle, with minor deviations in the low-frequency region. The axis scales of the graph are deliberately made equal size to check the ideality of the semicircle, which is frequently observed and usually expected in the case of electrochemical systems.

The equivalent electrical circuit that fitted all the experimental spectra is shown in the inset of Figure 10. It describes the charge transport through the current-limiting electrode-solution interface. The cell includes two electrodes, and one could expect the appearance of two capacitance-resistance blocks in the model. However, in the case of a relatively high current, the limiting effect of one photoanode becomes crucial, thus the impedance of the counter electrode seems to be negligibly small to have an impact on total system impedance. Therefore, the effect of only a photosensitive semiconductor-solution interface could be revealed from impedance spectrum analysis.

Elements R_1_ and C_1_ correspond to a charge transport process that was limiting the DSSC operation within the low voltage range, with resistance R_1_ being a charge transfer resistance of the cell. The physical sense of C_1_ element is better described as a capacitance of the electrode-electrolyte interface. The calculated values of all four parameters for all five types of cells are shown in Table 4.

The resistance R_1_ is considered the main parameter responsible for charge transfer rate through the electrode-solution interface. The values of its inverse value (charge transfer conductance) are compared in Figure 11.

The increase of R^−1^ value under illumination proves the photosensitivity of the electrode. The highest charge transfer conductivity (the lowest resistance) under light illumination was observed for the cell containing 1a active component (Figure 11). The charge transfer conductivity of the 2a based cells was smaller than the 1a based cells (Figure 11), however, the relative increase of conductivity under illumination was still high, especially in the case of 2a + N719 + CDCA sensitizer. The values of charge transfer resistance do not have to correspond directly to the photoconversion efficiency of the dye, which is a complex characteristic affected by many factors. The R value is not a photochemical, but an electrochemical parameter that describes the rate of charge transfer through the electrode-solution interface and is regarded to have the highest impact on the total internal resistance of the cell.

## 4. Conclusions

Two new benzo[*h*]quinolin-10-ol derivatives with one or two cyanoacrylic acid units were synthesized and characterized as well as tested as (co)-sensitizers DSSCs. Moreover, the beneficial impact of CDCA as a co-adsorbent was explained based on DFT calculations.

Summarizing the findings concerning the effect of the chemical structure of the prepared compounds, it was found that the introduction of a second anchoring group to benzo[*h*]quinolin-10-ol derivatives (i) significantly lowered melting temperature by about 65 °C but only slightly affected T_g,_ as well as its produced steel which is high at about 120 °C, (ii) lowered oxidation potential, and based on DFT in s **2a**@TiO_2_ system, the dye orbitals have a certain share (14%) in the HOMO level. This is contrary to **1a**@TiO_2_, where HOMO is located entirely on the dye (99%), and (iii) increased the molar absorption coefficient, slightly bathochromically shifted λ_max_, and when adsorbed in TiO_2_ showed a higher absorption range compared to **1a**@TiO_2_.

The DFT calculations of the utilization effect of three co-adsorbents, CDCA, CA and CDA, revealed that due to the almost perpendicular adsorption of the CDCA molecule and a higher value of *G*_ads_, CDCA seems to be a better co-adsorbent than others.

Considering the PV efficiency of fabricated solar cells based on synthesized molecules as co-sensitizers with N719, we improved the PCE value of about 12% (**1a**) and 19% (**2a**) with respect to the reference device based on a neat N719. The addition of CDCA increased the PCE of a reference cell from 5.35 to 6.90%. While the device with photoanode consists of a mixture of **2a**, N719 and CDCA showed a further increase of efficiency to 7.22%. Thus, the utilization of benzo[h]quinolin-10-ol derivative with two 2-cyanoacrylic acid units reduces the applied amount of N719 and enhances the DSSC performance.

## Figures and Tables

**Figure 1 materials-14-03386-f001:**
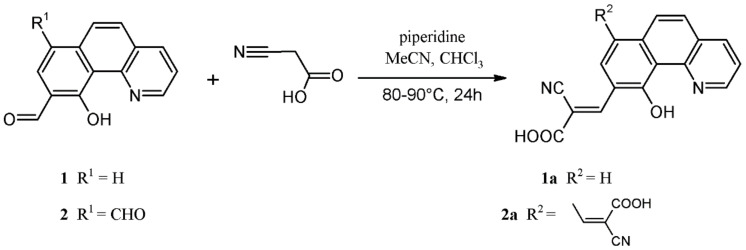
Synthetic route for the preparation of benzo[*h*]quinolin-10-ol derivatives **1a** and **2a**.

**Figure 2 materials-14-03386-f002:**
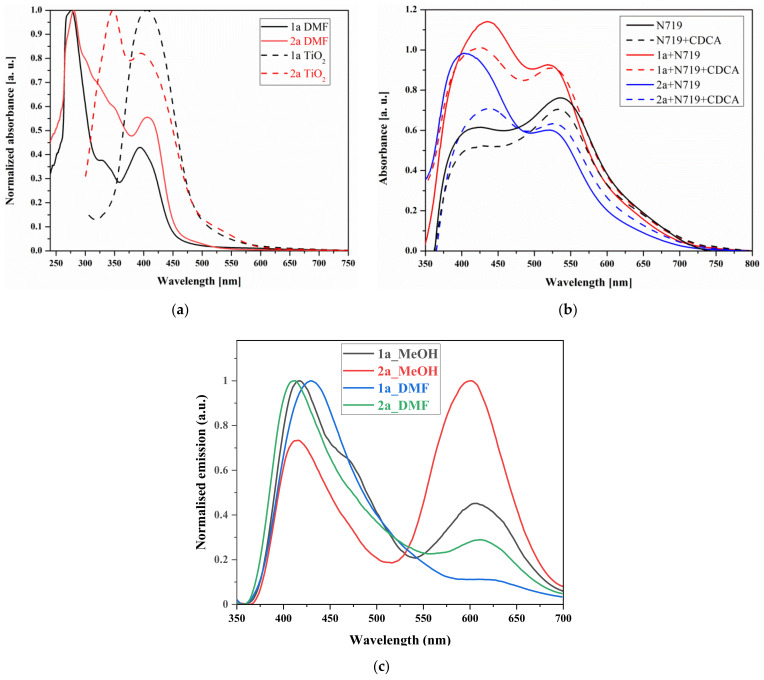
(**a**) UV-vis spectra of **1a** and **2a** in solution and TiO_2_ films with adsorbed dyes on glass substrates, (**b**) **1a** and **2a** as co-sensitizers with N719 with or without CDCA addition and (**c**) photoluminescence spectra of **1a** and **2a** recorded in the MeOH and DMF solutions (c = 10^–5^ mol/L).

**Figure 3 materials-14-03386-f003:**
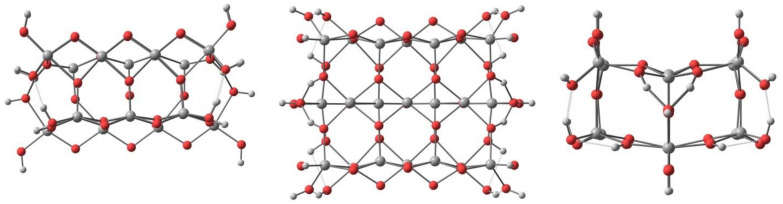
The top, front and side view of the Ti_21_O_54_H_24_ cluster.

**Figure 4 materials-14-03386-f004:**
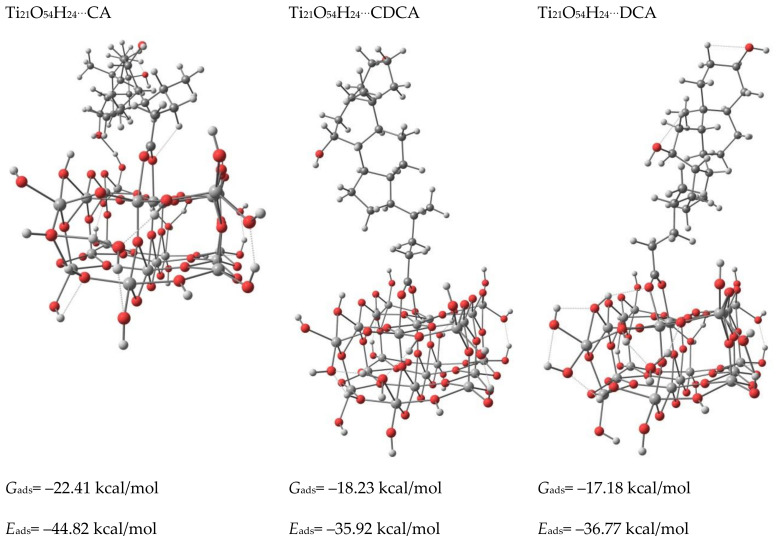
Adsorbed *keto*- forms of the dyes on Ti_21_O_54_H_24_ cluster (The Gibbs free energies G of the Scheme 298.15 K).

**Figure 5 materials-14-03386-f005:**
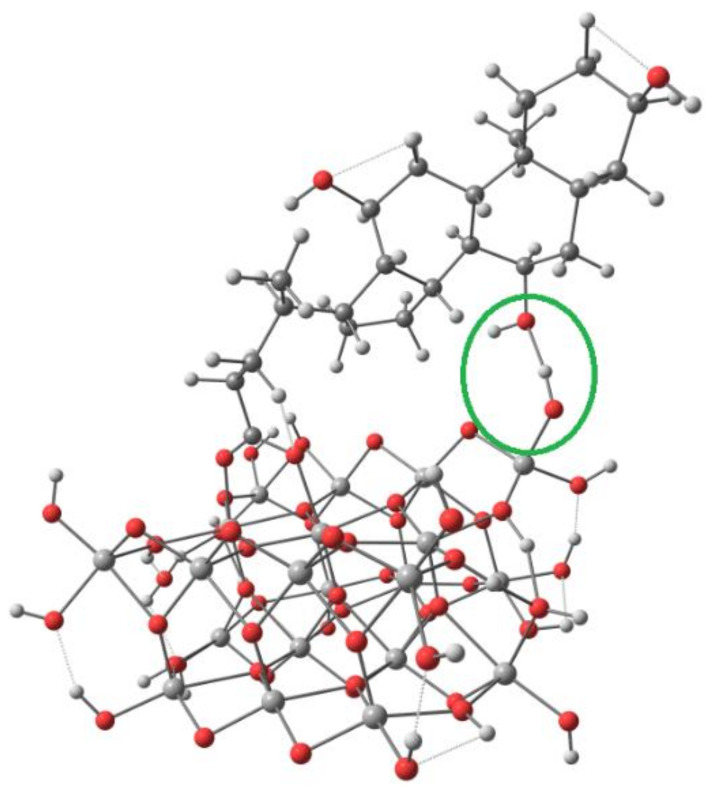
Hydrogen bond in the Ti_21_O_54_H_24_⋯CA system.

**Figure 6 materials-14-03386-f006:**
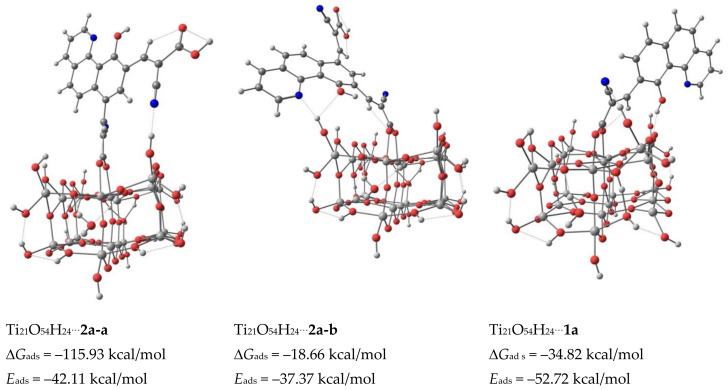
Adsorption of **1a** and **2a** on Ti_21_O_54_H_24_ clusters (Gibbs free energies *G* of the species were calculated at 298.15 K).

**Figure 7 materials-14-03386-f007:**
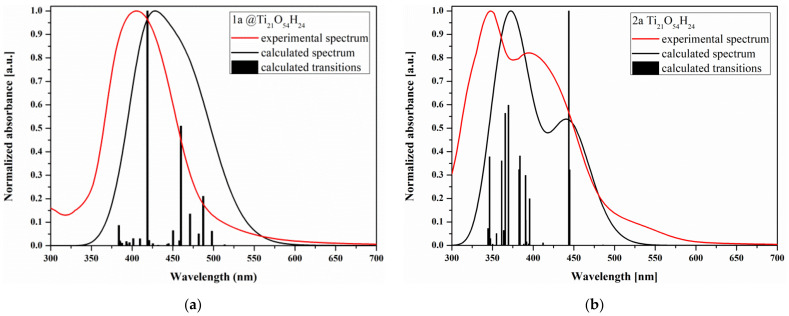
Normalized calculated and experimental UV-vis spectra of (**a**) **1a** or (**b**) **2a**@Ti_21_O_54_H_24_ systems.

**Figure 8 materials-14-03386-f008:**
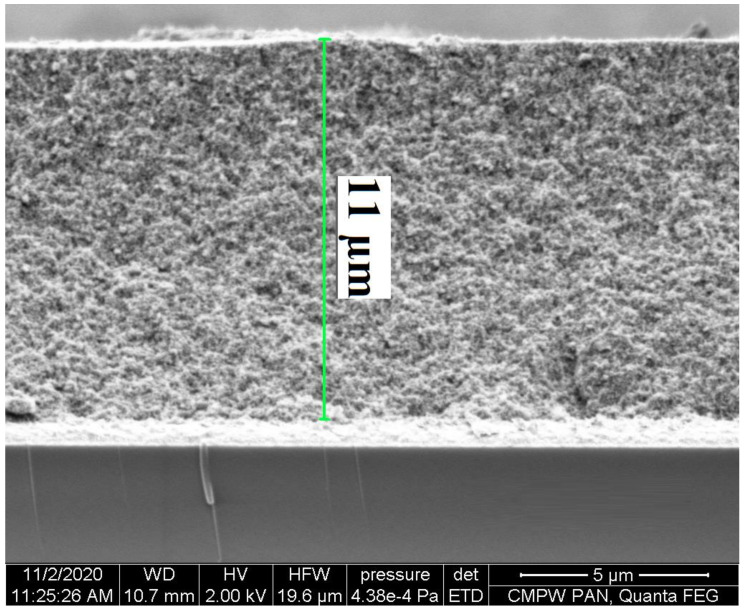
The cross-sectional SEM image of prepared TiO_2_ substrate with anchored dyes mixture **2a** + N719 + CDCA.

**Figure 9 materials-14-03386-f009:**
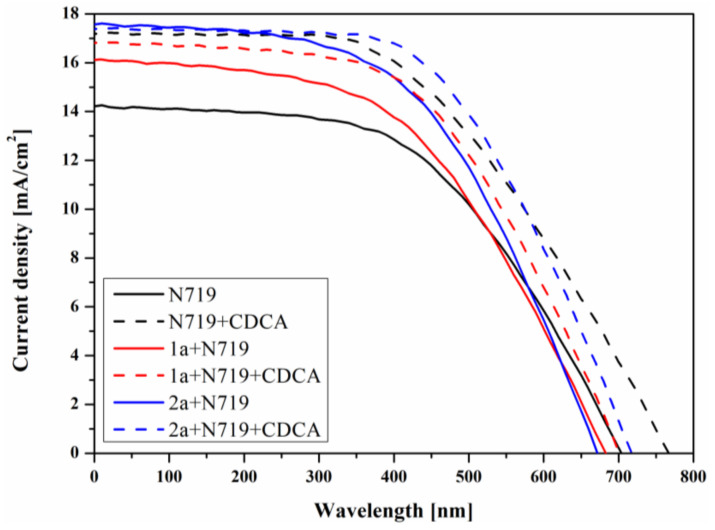
Photocurrent-voltage curves of prepared dye-sensitized solar cells.

**Figure 10 materials-14-03386-f010:**
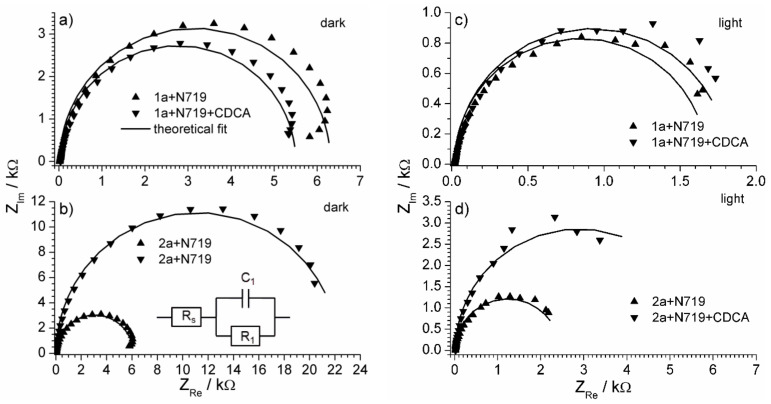
Impedance spectra (complex plane plots) of the studied cells: in darkness (**a**,**b**) and under light illumination (**c**,**d**). The photoanode dye component is shown in the legend. The equivalent electrical circuit is shown in the inset of part b.

**Figure 11 materials-14-03386-f011:**
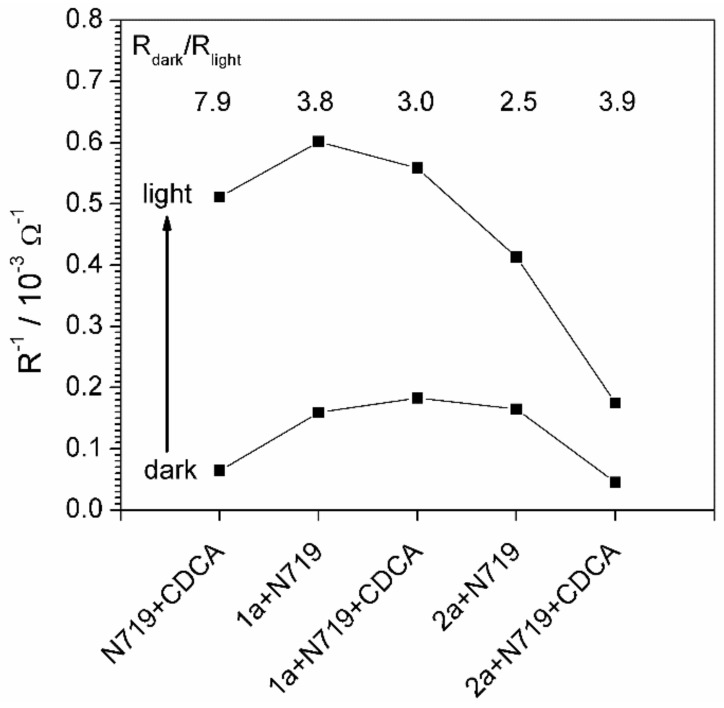
Inverse values of the charge transfer resistance compared for a set of cells in dark and under illumination. The calculated numerical increase of conductivity is given in the inscription.

**Table 1 materials-14-03386-t001:** Photophysical data for **1a** and **2a** recorded in MeOH and DMF solutions.

Solvent	λ_max_ (nm) (ε (M^−1^cm^−1^))		PL λ_em_ (nm)		τ_eff_ (τ (ns) (Weight%))		Φ (%)		E_g_^OPT^ (eV)	
	1a	2a	1a	2a	1a	2a	1a	2a	1a	2a
MeOH	275(10,880), 327(4635), 393(5085)	273(23,896), 366(13,600), 383 (12,727)	418, 611	413, 601	5.56 [1.65 (21.24), 6.62(78.76)]	5.97 [1.71(12.94), 6.60(87.06)]	0.07	0.15	2.97, 2.03	3.00, 2.06
DMF	276(16,000), 328(6000), 391(6667)	289(10,000), 354(10,667), 407(10,000)	430, 626	411, 610	4.24 [1.19(39.9), 6.26(60.1)]	4.57 [1.17(45.04), 7.35(54.96)]	1.13	1.25	2.89	3.02, 2.03

**Table 2 materials-14-03386-t002:** Characterization of the prepared photoanodes based on SEM, AFM and optical microscope measurements.

Photoanode	AFM	SEM	Optical Microscope		
	RMS (nm)	Thickness (µm)	Thickness (µm)	S_q_ (µm)	S_ku_
TiO_2_ + 1a	32	9	8	0.459	4.159
TiO_2_ + 2a	33	10	10	0.319	4.707
TiO_2_ + 1a + N719	29	8	9	0.383	4.331
TiO_2_ + 2a + N719	31	11	10	0.311	4.924
TiO_2_ + 1a + N719 + CDCA	35	12	12	0.598	5.479
TiO_2_ + 2a + N719 + CDCA	37	11	12	0.635	5.654

**Table 3 materials-14-03386-t003:** Photovoltaic parameters of fabricated dye-sensitized solar cells.

Photoanode Sensitized with	V_oc_ (V)	J_sc_ (mA cm^−2^)	FF (–)	PCE (%)
N719	0.703 (±0.004)	14.28 (±0.03)	0.53 (±0.02)	5.35 (±0.04)
N719 + CDCA	0.762 (±0.003)	17.20 (±0.05)	0.50 (±0.01)	6.90 (±0.07)
1a	0.475 (±0.006)	0.30 (±0.03)	0.49 (±0.02)	0.07 (±0.01)
2a	0.567 (±0.002)	1.38 (±0.02)	0.57 (±0.01)	0.44 (±0.02)
1a + N719	0.680 (±0.002)	16.12 (±0.01)	0.51 (±0.01)	6.02 (±0.03)
2a + N719	0.670 (±0.002)	17.57 (±0.03)	0.53 (±0.01)	6.38 (±0.04)
1a + N719 + CDCA	0.700 (±0.003)	16.94 (±0.08)	0.53 (±0.01)	6.42 (±0.08)
2a + N719 + CDCA	0.716 (±0.001)	17.44 (±0.04)	0.57 (±0.01)	7.22 (±0.04)

**Table 4 materials-14-03386-t004:** Calculated values of the equivalent circuit parameters.

	Dark			Light		
Compound	R_s_/Ω	R_1_/kΩ	C_1_/μF	R_s_/Ω	R_1_/kΩ	C_1_/μF
N719 + CDCA	16.0	15.5	16.1	31.1	1.96	21.5
**1a** + N719	19.9	6.28	18.0	16.7	1.66	19.1
**2a** + N719	18.8	6.07	18.4	17.3	2.42	20.4
**1a** + N719 + CDCA	37.8	5.45	19.1	22.3	1.79	21.6
**2a** + N719 + CDCA	21.6	2.22	16.2	19.7	5.72	18.6

## Data Availability

The data presented in this study are available on request from the corresponding author.

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
