# Peer review of "New Benzo[h]quinolin-10-ol Derivatives as Co-sensitizers for DSSCs"

_materials, 2021, doi:10.3390/ma14123386_

Round 1

Reviewer 1 Report

Manuscript ID: materials-1238556

Title: New Benzo[h]quinolin-10-ol Derivatives as Co-sensitizers for  DSSCs

Author: Slodek et. al.

In this work, the authors synthesized benzo[h]quinolin-10-ol derivatives with one or two 2-cyanoacrylic acid units with good yield in a one-step condensation reaction and characterized using NMR spectroscopy and elemental analysis. Photophysical properties of the DSSC with the use of co-sensitizer investigated by series of experiments such as differential scanning calorimetry, cyclic voltammetry, electronic absorption and photoluminescence. Also, the authors used DFT analysis to explore the structure-activity relationship of co-sensitizers and the efficiency of DSSC. The manuscript was written in a scattered way, especially the computational part. Also, the use of computational methods is not well justified and used discrepantly. There are some typos like TF-DFT. I do not agree to accept this article in the current version.

Comment1. 

I think the last line in the abstract is quite overly stated “For the first time, the effect of co-adsorbent chemical structure …. explain based on the density functional theory.” So far, I remember at least two works related to the use of DFT to explore the effect of structural modification of co-sensitizers and efficiency. 

  1. DOI: 10.1016/j.solener.2017.02.046
  2. DOI: 10.1021/acs.jpcc.8b06495

Comment2.

The authors used different functionals (BP86 and B3LYP) for two different systems. I believe the author can use B3LYP functional for all the systems considered in the study. If we compare the computational cost, both will take similar resources.   

Comment3.

In literature, it is well established that the anatase(101) surface is most stable and mostly used for computational modeling. Is there any particular purpose to use a rutile (110) surface?

Comment4.

In section 3.2.2, the author mentioned the dye/cluster geometry, but I did not see any optimized system structure in the main manuscript and supplementary. It will increase the impact of this work if the author can add the optimized structure of the system(Ru-dye/Co-sensitizer/TiO2).

Author Response

Dear Reviewer

We thank you for your detailed comments on our manuscript materials-1238556 entitled “New benzo[h]quinolin-10-ol derivatives as co-sensitizers for DSSCs”. We have carefully considered yours comments. Modifications and corrections have been added accordingly in the reply to yours suggestions.

In this work, the authors synthesized benzo[h]quinolin-10-ol derivatives with one or two 2-cyanoacrylic acid units with good yield in a one-step condensation reaction and characterized using NMR spectroscopy and elemental analysis. Photophysical properties of the DSSC with the use of co-sensitizer investigated by series of experiments such as differential scanning calorimetry, cyclic voltammetry, electronic absorption and photoluminescence. Also, the authors used DFT analysis to explore the structure-activity relationship of co-sensitizers and the efficiency of DSSC. The manuscript was written in a scattered way, especially the computational part. Also, the use of computational methods is not well justified and used discrepantly. There are some typos like TF-DFT. I do not agree to accept this article in the current version.

Comment1. 

I think the last line in the abstract is quite overly stated “For the first time, the effect of co-adsorbent chemical structure …. explain based on the density functional theory.” So far, I remember at least two works related to the use of DFT to explore the effect of structural modification of co-sensitizers and efficiency. 

  1. DOI: 10.1016/j.solener.2017.02.046
  2. DOI: 10.1021/acs.jpcc.8b06495

Answer: Thank you for remark. Considering the mentioned by Reviewer work DOI: 10.1016/j.solener.2017.02.046 the effect of co-adsorbent has not been presented in this paper. In the second mentioned by Reviewer work DOI: 10.1021/acs.jpcc.8b06495, indeed the effect of co-adsorbent (chenodeoxycholic acid) on device performance was explain based on DFT. We are sorry for omission of this important work, which we has been cited as ref.[61] in the revised version of our paper. In our work we compare the effect of chemical structure of three co-adsorbents, that is, cholic acid, deoxycholic acid  and chenodeoxycholic acid, not only chenodeoxycholic acid as takes place in [61]. According, to the research given in [61], the sentences form Abstract, Introduction and section 3.2.1. have been modified as follow:

Abstract: “Additionally, the effect of co-adsorbent chemical structure (cholic acid, deoxycholic acid  and chenodeoxycholic acid) on DSSC efficiency was explain based on the density functional theory.”

Introduction:Moreover, the impact of the chemical structure of typically used adsorbents, such as chenodeoxycholic acid (CDCA), cholic acid (CA), and deoxycholic acid (DCA) on device performance based on N719 was explained by DFT.

Section 3.2.1.: “Moreover, G. Saranya et al. [61] based on DFT, showed that CDCA co-adsorbent is a crucial component of a high-performance DSSC. They studied effect of CDCA on the properties of the dye denoted as TY6’ and TiO2 interface. It was found that, CDCA not only stabilizes the TY6’/TiO2 system but also prevents the surface tensile stress induced by the dye monolayer [61].”

Comment2.

The authors used different functionals (BP86 and B3LYP) for two different systems. I believe the author can use B3LYP functional for all the systems considered in the study. If we compare the computational cost, both will take similar resources.   

Answer: Thank you for remark. We would like to explain that the calculations were performed using B3LYP functional. The error in the text has been corrected.

Comment3.

In literature, it is well established that the anatase(101) surface is most stable and mostly used for computational modeling. Is there any particular purpose to use a rutile (110) surface?

Answer: The choice of rutile (110) surface was not dictated by any special consideration. We would like to explain that the purpose of our calculations was to determine the relative values of the thermodynamic parameters of the adsorption of CA, DCA, CDCA, 1a and 2a dyes on the TiO2 surface for comparison of the effect of the chemical structure co-adsorbents and sensitizes. In our opinion effect of the choice of rutile (110) or anatase (101) does not have key importance in the case if the study concerns estimation of relative values of adsorption energy compounds to the TiO2. Thus, it is of minor significance if for surface simulation rutile or anatase structure is applied and is of no importance which lattice plane of the crystal will be selected [Applied Surface Science 428 (2018) 118–123]. The surface of semiconductor do not consist of perfectly arranged TiO2 crystals. It can be supported for the fact that there are papers in which such information concerns of TiO2 type for computational modelling is not given [Materials Today Communications 22 (2020) 100731; Optik - International Journal for Light and Electron Optics 208 (2020) 163920; Journal of Materials Research and Technology 9 (2020) 1175].

Comment4.

In section 3.2.2, the author mentioned the dye/cluster geometry, but I did not see any optimized system structure in the main manuscript and supplementary. It will increase the impact of this work if the author can add the optimized structure of the system(Ru-dye/Co-sensitizer/TiO2).

Answer: According to Reviewer remark Cartesian coordinates of optimized geometries of the studied dye@Ti21O54H24 systems have been presented in Table S7 in the supplementary materials.

Reviewer 2 Report

The contents are interesting however, please try to establish the need for this scheme.

Define your experimental details - line 78

on the computational studies, a compound/chemical reaction formula should be presented.

Additional interpretation on Figure 8.

Define your major contribution in your conclusion

Provide latest references within 3 years

Author Response

Dear Reviewer

We thank you for your detailed comments on our manuscript materials-1238556 entitled “New benzo[h]quinolin-10-ol derivatives as co-sensitizers for DSSCs”. We have carefully considered yours comments. Modifications and corrections have been added accordingly in the reply to yours suggestions.

Remark 1. The contents are interesting however, please try to establish the need for this scheme.

Answer: We afraid if we not sure what Reviewer remark means.

The aim of our work is focused on looking for new materials for applications in dyes sensitized solar cells (DSSCs). In the paper we present the results of our investigations in typical scheme for chemical articles. Thus, the work is divided in Introduction, Experimental, Results and Discussion and Conclusion parts. in Experimental section the synthesis, chemical name of compounds and its structural characterization are given. Results and Discussion consist of few section in which synthesis of designed compounds is described and in next parts the selected physical properties supported by DFT calculations are presented. The structure of the compounds applied as photosensitizer, which harvests photons and initiates the electrochemical process, plays a significant role in the overall performance of the cell. Thus, the basic knowledge concerns relationship between structural elements of compounds and the most important properties from the point of view DSSC has significant importance. Finally photovoltaic characterization carried out by current-voltage and electrochemical impedance spectroscopy measurements of fabricated solar cells based on synthesized benzo[h]quinolin-10-ol derivatives are showed. Due to the utilization of synthesized benzo[h]quinolin-10-ol derivatives as co-sensitizers, the better photovoltaic performance of fabricated devices compared to a reference cell based on a neat N719 was demonstrated.

Remark 2. Define your experimental details - line 78 on the computational studies, a compound/chemical reaction formula should be presented.

Answer: Line 78 it is section 2. Experimental. According to Reviewer remark to Experimental the section 2.1. Computational details has been added.

“2.1. Computational details

All theoretical calculation were performed using Gaussian 16, Revision C.01, program package [43] at DFT or TD-DFT level. The singlet state geometry optimizations, frequency and electronic transition calculations were made with the use of B3LYP functional [44-45] with the 6-31G(d,p) basis set [46]. Calculations were made in gas phase except for the absorption electronic spectra of 1a, 2a compounds (Figure S3) in addition to the PCM model [47] with DMF as solvent was used. The density-of -states diagrams (Figure S13) were obtained with GaussSum [48]. The adsorption energies (Eads) of the dyes have been evaluated using the following expression, Eads = Edye + ETiO2 − Edye@TiO2, where, Edye, ETiO2 and Edye@TiO2 are the energies of dye, TiO2, and total

system (dye@TiO2). The Gibbs free energies of the dye@TiO2 species were calculated at 298.15K.”

A compound/chemical reaction formula is presented in Figure 1 in section 3. Results and discussion. As usually in chemical papers, in Experimental section the synthesis, chemical name of compounds and its structural characterization are given. Figure 1 presents synthetic route for the preparation of benzo[h]quinolin-10-ol derivatives and its chemical structure

Remark 3. Additional interpretation on Figure 8.

Answer: Figure 8. presents the cross-sectional scanning electron microscope (SEM) image of photoanode, that is glass with FTO covered with TiO2 with anchored dyes mixture 2a+N719. Additional images of the cross-sections of fabricated photoanodes are showed in Figure S14. Such SEM images give information about thickness of the TiO2 layer. The thickness of the mesoporous oxide layer was determined, as it greatly affects the optical properties of the photoanodes and prepared solar cells photovoltaic performance [45]. In the presented work the thickness of the TiO2 was investigated using two methods, that is, optical microscope and SEM. The advantage of the SEM microscope over the optical microscope is that the thickness of individual layers can be determined. On the SEM images the layers of glass, FTO and mesoporous TiO2 were seen. However, that thickness determination using an optical microscope is a non-destructive method. Considering the data given in Table 2, it can be concluded that only small differences in the obtained thickness values of TiO2 were observed using two independent measuring methods (SEM and optical microscope).

We proposed to add the following additional interpretation on Figure 8 and Figure S14 to the revised version of our paper:

“In the presented work the thickness of the TiO2 was measured using two methods, that is, optical and SEM microscopes. The advantage of the SEM over the optical microscope is that the thickness of individual layers can be determined. However, that thickness determination using an optical microscope is a non-destructive method. The representative cross-sectional SEM image and micrograph of photoanodes are depicted in Fig. 8 and Fig. S14.  On the SEM images the layer of glass, FTO and mesoporous TiO2 were seen (cf. Fig. 8 and Fig. S14). Moreover, based on SEM, the presence of spherical TiO2 nanoparticles was confirmed. Considering the data given in Table 2, it can be concluded that only small differences in the obtained thickness values of TiO2 were observed using two independent measuring methods (SEM and optical microscope).”

Remark 4. Define your major contribution in your conclusion

Answer: The major contribution in our conclusions is connected with design, and synthesis of benzo[h]quinolin-10-ol derivative, which applied as co-sensitizer let to   enhances the DSSC performance and reduces the amount of N719.

Remark 5. Provide latest references within 3 years

Answer: We would like to explain that the latest references within 3 years have been presented in references list. References list contains 65 positions among which 25 have been published in 2018, that is, within 3 years.

Round 2

Reviewer 2 Report

All necessary comments last time were enhanced to provide a better manuscript.

  1. minimal enhancement of images/figure needs to be changed especially figure 7. 
  2. moreover, enhancement of a better figure 8 to provide a visible details on it

Author Response

Dear Reviewer,

Thank you very much for the comments you sent, we are glad that the changes previously made have improved the quality of our manuscript. 

According to the current recommendations, we have improved the quality and readability of all figures.

Best regards